# Recent Progress of Metal-Organic Frameworks and Metal-Organic Frameworks-Based Heterostructures as Photocatalysts

**DOI:** 10.3390/nano12162820

**Published:** 2022-08-17

**Authors:** Mohammad Mansoob Khan, Ashmalina Rahman, Shaidatul Najihah Matussin

**Affiliations:** Chemical Sciences, Faculty of Science, Universiti Brunei Darussalam, Jalan Tungku Link, Gadong BE 1410, Brunei

**Keywords:** metal-organic frameworks, MOF, photocatalysis, organic linkers

## Abstract

In the field of photocatalysis, metal-organic frameworks (MOFs) have drawn a lot of attention. MOFs have a number of advantages over conventional semiconductors, including high specific surface area, large number of active sites, and an easily tunable porous structure. In this perspective review, different synthesis methods used to prepare MOFs and MOFs-based heterostructures have been discussed. Apart from this, the application of MOFs and MOFs-based heterostructures as photocatalysts for photocatalytic degradation of different types of pollutants have been compiled. This paper also highlights the different strategies that have been developed to modify and regulate pristine MOFs for improved photocatalytic performance. The MOFs modifications may result in better visible light absorption, effective photo-generated charge carriers (e^−^/h^+^), separation and transfer as well as improved recyclability. Despite that, there are still many obstacles and challenges that need to be addressed. In order to meet the requirements of using MOFs and MOFs-based heterostructures in photocatalysis for low-cost practical applications, future development and prospects have also been discussed.

## 1. Introduction

Photocatalysis is a chemical reaction that takes place in presence of a photocatalyst and a suitable light. In short, it is the acceleration of a photo-reaction in the presence of a catalyst. Photocatalysis over semiconductors occurs when light with an energy greater than the band gap of semiconductors is absorbed generating photo-excited electrons (e^−^) and holes (h^+^) for subsequent reduction and oxidation processes by e^−^ and h^+^, respectively [1]. Metal oxides and chalcogenides are two semiconductors that have been extensively studied. Metal oxides, such as TiO_2_ [2], ZnO [3], SnO_2_ [4], and CeO_2_ [5,6], are commonly used as photocatalysts. Despite their stability, the main drawback of using metal oxides as photocatalysts is their wide band gap energy. The wide band gap only enables them to absorb ultraviolet light and this restricts their practical usage [7]. In addition, carbon-based materials, such as graphene and g-C_3_N_4_, have also attracted remarkable research interest in recent years, owing to their ultrathin, two-dimensional nature, and unique properties including high thermal conductivity and excellent mechanical stiffness [8,9]. However, the fast recombination of the charge carriers is the major disadvantage of using graphene as a photocatalyst material. To overcome this problem, it requires modification with nanoparticles or light-sensitive materials. In contrast, chalcogenides exhibit narrow band gap energy. However, it has the disadvantage of having a high rate of recombination of photo-generated ē and h^+^ as well as the photo-corrosion phenomena [10,11].

Recently, metal-organic frameworks (MOFs) have emerged as novel photocatalysts due to their intrinsic structural properties by having a high surface area and a well-ordered porous structure [12,13]. MOFs are porous crystalline hybrid materials formed by the linkage of metal centers (cluster) and organic ligands (organic linkers). These MOFs are known as porous coordination polymers [14]. MOFs exhibit reticular design in which two designs approach: supermolecular building layer and supermolecular building block demonstrate the influential capability of the MOFs as precursor materials with unique structure. In general, metal-enclosed inorganic compounds and organic ligands (moieties) are linked through coordination bonds, thus developing MOFs. Therefore, a three-dimensional interconnected network is formed creating open frameworks with high permanent porosity [15].

Properties such as regular channels, high specific area, and abundant active sites make MOFs as promising materials [16]. MOFs are considered low electron conductivity materials due to the poor overlap between frontier orbits and the electronic states on ligands and metal ions. However, the electronic properties of a metal can be changed by changing its coordination environment to enforce the desired geometries around the metal [17]. Figure 1 shows some of the tunable properties of MOFs [18]. In the last decade, MOFs have been employed for water treatment [19], chemical hydrogen storage [20], photochemical and electrochemical energy storage and conversion [21,22], removal of heavy metals [23], solar energy conversion [24,25], fuel production [26], gaseous fuel storage [27], water harvester [28], water splitting for hydrogen evolution reaction [29], and oxygen evolution reaction [30]. The organic linkers tend to have large π-conjugated systems in which electron transfer between ligands and metals is possible. Interestingly, if the MOFs have a metal-ligand with photocatalytic properties, then ligands themselves can generate photo-generated electrons for photocatalytic reactions [31].

Therefore, the advantages of using MOFs as photocatalytic materials are as follows [26]:The porous structure of MOFs exposes more active sites and facilitates;Ordered crystal structure of MOFs can separate electron-hole pairs;MOFs structure can be tailored;MOFs are easy to separate and recycle from the reaction system;MOFs have the tendency for electron transfer between ligands and metals due to the large π-conjugated systems in the organic linkers;A long-wavelength absorbing organic bridging ligand can enhance the optical absorption of MOFs.

MOFs are widely researched due to their potential applications such as photocatalytic of degradation of organic and inorganic pollutants [32,33], photocatalytic of antibiotics [34], photocatalytic antimicrobials [35], bio-imaging [36], drug delivery [37], photocatalytic organic transformations [36,38], photocatalytic production of hydrogen (H_2_) [39], and photocatalytic reduction of carbon dioxide (CO_2_) [40], as illustrated in Figure 2.

However, recombination of photogenerated electron-hole pairs has become the main challenge that limits the improvement of photocatalysis [41]. By constructing heterostructure, the charge separation can be promoted due to difference in the interface potential that generates a built-in electric field in which it helps in inhibiting recombination [42]. A typical strategy includes the introduction of metal nanoparticles to improve the separation of photogenerated electron-hole pairs [41]. Another strategy is to integrate MOFs and inorganic semiconductors/organic polymers. The interaction between the inorganic-organic complex is versatile due to the strong charge interaction between components [41]. In addition, the organic linkers of MOFs could also be further modified which have a great potential to construct heterostructures [43]. Therefore, MOFs-based heterostructure photocatalysts are mainly composed by different components.

In summary, this article emphasizes the recent progress and developments in the photocatalytic activities of MOFs and MOFs-based heterostructures. Various synthesis methods to fabricate MOFs and their respective heterostructures have been discussed in-depth. Moreover, the possible mechanisms of the photocatalytic performance of MOFs and MOFs-based heterostructures photocatalysts have also been illustrated and discussed. Despite the current progress of MOFs and MOFs-based heterostructures, there are still limitations and gaps that hinder the potential of the materials in different applications. Therefore, the future prospects and conclusions have also been highlighted and discussed.

## 2. Different Synthesis Methods of MOFs and MOFs-Based Heterostructures

### 2.1. Synthesis of MOFs

The main goal in MOFs synthesis is to establish the synthesis conditions that lead to defined inorganic building blocks without the decomposition of the organic linker. The understanding of typical metal coordination environments or the formation conditions of typical inorganic building blocks is crucial for direct synthesis efforts [44]. Higher surface area (ultrahigh porosity) of MOFs provides extraordinary properties. This requires an increase in storage space per weight of the material. The pore size and topology of the framework can be finely tuned by selecting appropriate linkers and metal nodes. Longer organic linkers provide larger storage space and a greater number of adsorption sites. Moreover, MOF topology has shown some effects on catalytic performance [45]. However, a unique spatial distribution of active sites can only be achieved within a specific MOF structure. Therefore, the topological diversity of MOFs can be used to change the position and orientation of MOF catalytic sites which can be located at the nodes or the linkers [46]. Ideally, it is important to keep the pore diameter below 2 nm in order to maximize the surface area of the framework [47]. Apart from that, MOFs are well-known for their ability to impart functionality using a judicious selection of linkers and metal nodes. They undergo post-synthetic modification to further tune properties through swapping or altering linker or node components in the framework. Furthermore, select crystallographic phases and crystallite size/morphology can be controlled to modify the surface chemistry of the MOF [18]. In the last decade, various synthesis methods have been used to design MOFs, such as hydrothermal [48], solvothermal [49], precipitation [50], and microwave-assisted synthesis [51].

#### 2.1.1. Microwave-Assisted Synthesis

Microwave synthesis [48,49,50,51] is one of the most common synthesis methods. It uses microwave irradiation, which facilitates the nucleation and crystal growth in the synthesis of MOFs synthesis [51]. For instance, mesoporous iron trimesate nano MOF MIL-100(Fe) was synthesized using microwave-assisted hydrothermal reaction [52]. In general, due to the high-frequency motion generated by microwave heating, the heating is uniform without the need for heat conduction. Apart from the temperature, the pressure of the reaction process can also be controlled. This method has a significant improvement in saving reaction time and reducing energy consumption [53]. This method obtained a homogenous and chemically stable coating of polyethylene glycol (PEG) with enhanced colloidal and chemical robustness. Moreover, it was found that the material induced a low immune response. [RE(L)(H_2_O)(DMF)_n_] where RE = Dy, Gd, Ho, Pr, and Sm have been synthesized through the hydrothermal method [54]. Structural analyses reveal that the ligands are linked by five different rare-earth centers forming iso-structural nanoporous frameworks. Other work by Goswami et al. reported on the synthesis of 2D-MOFs with redox active metal centers (Ni(II), Co(II), and Cu(II)) through hydrothermal synthesis [55]. The metal centers contain two types of electron-rich linkers such as bis(5-azabenzimidazole, Linear L_1_, and angular L_2_) and aromatic dicarboxylates. In another study, Zr-based metal-organic frameworks were prepared using microwave-assisted synthesis [51]. The porosity of the material was based on the quantity of the modulator, reaction time, and temperature. It was found the optimum amount of modulator was in the range of 40 to 185 mol and the optimum temperature was between 80 to 120 °C.

#### 2.1.2. Solvothermal Method

MOFs have also been synthesized using the solvothermal method as it is considered one of the most popular methods used to prepare MOFs. The solvothermal method offers controllable geometry of the synthesized product with a large yield compared with other methods [56]. Solvothermal reactions are carried out in the liquid phase and crystals are precipitated through self-assembly between organic ligands and metal salts to obtain target MOFs [53]. Denisov et al. reported the synthesis of [Zn_4_O(BDS)_3_] via the solvothermal method [49]. The Zn_4_O^6+^ tetrahedral are bound to each other into a three-dimensional cubic framework by the BDC^−^ anions to form large spherical pores with a volume of 1838.78 Å^3^ accessible for the solvent. A facile and template-free solvothermal method has been used to prepare Cu-based MOFs nanosheets as reported by Zhang et al [57]. The starting coordination complexes of different copper (II)-ligand was found to control the growth and morphology of MOF crystal. Duan et al. synthesized Cu-BTC using a room temperature-template strategy [50]. The synthesized material exhibited a hierarchically porous structure and excellent thermostability. Moreover, various organic solvents, such as dimethyl sulfoxide, N, N-dimethylformamide, and toluene, have been widely used to fabricate MOFs and MOFs-based heterostructures to date. Among the above-mentioned organic solvents, dimethylformamide is most often selected for the synthesis of MOF and its modifications, due to its high boiling point and solubility. Although the solvothermal method has been widely used, the toxicity and environmental unfriendliness of the organic solvents severely restrict their use.

#### 2.1.3. Vapor Assisted Synthesis

In Virmani’s work, UiO-66 MOFs were obtained from a vapor-assisted conversion synthesis [58]. A mixed solution of ZrOCl_2_, dicarboxylic acid, and acetic acid were prepared and deposited. The deposited materials were stored in an oven and dried under pressure subsequently. The thickness of the MOF films was found to be 200 nm. A copper dicarboxylate metal-organic framework film was prepared by chemical vapor deposition (CVD) as stated by Stassin et al. [59]. The MOF-CVD method involved two steps. Firstly, the substrate was changed into a Cu precursor layer by physical vapor deposition and secondly, the precursor layer reacted with the vapor of the dicarboxylic acid linker. The authors extended this process to a thin film of copper dicarboxylate MOFs, which was the first crystalline and oriented porous material to be obtained entirely through vapor deposition. In another work, high quality Mg-MOF thin films were prepared using vapor assisted synthesis [60]. A mixture of H_2_O, acetic acid, and dimethylformamide (DMF) was used as the vapor source and filled into a sealed glass container. The precursor, Mg(CH_3_COO)_2_·4H_2_O was dissolved in DMF and 2,5-dihydroxy-benzenedicarboxylic (DOBDC) acid in DMF. AA was added into the precursor solution to prevent a rapid precipitation of the precursors when the Mg salt and the DOBDC mixed.

#### 2.1.4. Sonochemical Method

Ultrasonic irradiation has also been used in the synthesis of MOFs. The energy of high frequency in the liquid phase induces the generation, growth, and rupture of bubbles which make the local instantaneous temperature and pressure of the reactant increase [53]. The immense energy and high pressure created during the sonication process help to create a stable structure. Thus, the sonication technique has a great potential for the fabrication of MOFs in large quantities and at a low cost. For instance, Zn-based MOFs were synthesized by the aforementioned method [61]. DMF and triethylamine were included in the reaction and were mixed with the precursor vigorously until the solution was clear. An ultrasound rod was then placed into the reaction reactor. The obtained product was filtered and obtained uniform nucleation and high production yield. Table 1 shows the various synthesis of MOF materials.

There are several works relied on low-temperature routes to synthesize MOFs. A precipitation reaction method is well-known to grow simple molecular or ionic crystals since it is possible to tune the reaction conditions [62]. Vapor phase synthesis has been used as it can gently and efficiently produce oriented MOF films [63]. Since high orientation and high crystallinity MOFs are produced, they are suitable for applications in sensor and electronic devices. Meanwhile, microwave-assisted provides an efficient and sustainable route to synthesize MOF materials. This is due to the microwave energy directly penetrating into the substance as opposed to just relying on the heat conduction of the medium for heating. The microwave-assisted method usually takes only a short amount of time to complete the entire heating process which can effectively reduce the temperature and time [64]. The sonochemical method on the other hand can improve the crystallization and growth rate during the growth process [65].

### 2.2. Synthesis Method of MOFs-Based Heterostructures

In addition to the synthesis of MOFs as mentioned earlier, the synthesis of MOFs-based heterostructures is also widely studied in the field of heterogeneous catalysis due to their large specific surface area, modifiable pore structure, and high exposed active sites [66]. The construction of heterojunction is an effective method to improve the photocatalytic property of photocatalyst by improving the photogenerated carrier separation efficiency [67,68]. Moreover, the porous nature of MOFs facilitates the migration and separation of photo-generated charge carriers. The following shows the different modifications employed to synthesize MOFs-based heterostructures.

#### 2.2.1. Addition of Linkers

The MOF structures may be easily altered by changing the substituent groups in linkers and/or linker itself, as well as exchanging or doping metal ions in the framework nodes [42,69]. As a result, the band gap energy of MOFs can be easily controlled to improve photocatalytic efficiency. For instance, Dan-Hardi et al. have synthesized highly porous titanium-based MOFs using terephthalic acid (H_2_BDC) precursor and they found out that MIL-125(Ti) synthesized with unmodified H_2_BDC can only respond to UV light. Another study, reported by Fu and co-workers introduced a NH_2_ group into the H_2_BDC precursor and successfully synthesized isostructural NH_2_-MIL-125(Ti) in the same way. The authors reported that the synthesized materials displayed yellow color and their light absorption ability extended to the visible region. The band gap dramatically decreased from 3.60 to 2.46 eV. Moreover, this shows that the incorporation of BDC derivatives could tune the optical properties of the MOFs [70].

The organic linkers are used in some modifications because they can function as an antenna for light sensitizing and charge transfer to the inorganic cluster by exploiting ligand-to-metal charge transfer (LMCT). The LMCT feature of MOFs is ideal due to the efficient charge separation. Moreover, alteration of the organic linker could introduce additional substituents (using mixed linkers or even capping additional metal ions) affecting the oxidative power of these MOFs. A similar concept may be used for the reduction; this is because metal orbitals in MOFs define the position of the lowest unoccupied crystal orbital (LUCO), and the reductive power can be changed by choosing metal ions with suitable orbitals [71]. For instance, Wang et al. have successfully fabricated porphyrin-MOFs to degrade bisphenol F under irradiation of visible light [72]. The organic linkers of porphyrin can harvest light by playing the role of antennas, and further activate the metal clusters via a linker to a cluster charge transition that could generate enough reactive oxygen species (ROS). They also reported that the prepared photocatalyst was quite stable in a saline environment and could achieve over 78% bisphenol F removal (with/without salt) after 8 cycles.

#### 2.2.2. Modification of its Metal Clusters

Alternatively, the optical response can be altered by tuning the cluster-forming metal or by employing mixed metal clusters. The latter approach has been utilized to create mid-gap metal-centered states that result in the MOFs with LMCT transition which are clusters that cannot be fabricated via direct synthesis. The second method is to use MOFs as a passive container for encapsulating a light-absorbing photocatalyst. These active species are often homogeneous catalysts based on precious metals, and leaching has been effectively prevented by encapsulating or covalently bonding them to the framework [73,74]. Table 2 shows some of the MOFs-based heterostructures that have been used in various photocatalytic activities.

MOFs have a number of advantages in photocatalysis due to their inherent structural features which include large surface area and porous structure and a tunable combination between metallic nodes and organic linkers. The photocatalytic efficiency is, however, still insufficient to meet the actual needs. As discussed above, a series of strategies have been introduced to enhance the photocatalytic properties of MOFs and their applications including ligand functionalization [87], mixed ligands [88], mixed metal centers [89,90], metal ions immobilization [91], the addition of magnetic materials, MOF or COF coupling [92,93], semiconductors coupling [94,95], surface decoration using carbon-based materials [95] or metal nanoparticles [96], and dye sensitization [40]. These strategies have also been employed to extend visible light absorption of MOFs, more efficient generation, and separation and transfer of charge carriers, as well as its good recyclability. Owing to its outstanding properties, all of the above-mentioned MOFs-based heterostructures can effectively use light energy for various photocatalytic activities, and most of them can be recycled several times. Due to their extraordinary porosity and large surface area, both the metal nodes and organic linkers can be easily tuned; thus, MOFs-based heterostructures exhibited promising photocatalytic performance [97]. However, the stability of MOFs during the photocatalytic reaction is an important issue needed to be further considered, thus searching MOFs with optimized coordination between metal clusters and organic linkers for high stability is of great importance.

## 3. Applications of MOFs and MOFs-Based Heterostructures

The use of MOFs as photocatalysts for photocatalytic activities has been reported in previous literature and discussed below. Table 3 summarizes the results of photocatalytic degradation of various waste reported in the different studies.

### 3.1. Photocatalytic Evolution of H_2_

Non-metal g-C_3_N_4_ is a well-known photocatalyst due to its appealing electronic structure, low cost, and high stability. The band gap of g-C_3_N_4_ was approximately 2.7 eV, indicating the light absorption edge into the visible region (up to 450 nm). Thus, coupling g-C_3_N_4_ with MOFs may lead to an enhanced visible light response. For instance, Zhao et al. investigated the photocatalytic evolution of H_2_ using protonated g-C_3_N_4_ co-operated with Sm-doped Co-MOF to form a 2D/2D heterojunction via electrostatic self-assembly [108]. The H_2_ evolution rate of 5% Sm-doped exhibited the highest activity has reached 73.42 μmol h^−1^ within 5 h under simulated sunlight. The improved activity may be attributed to the presence of a small amount of Sm ions in the trivalent state which increases the chaos of coordination in the Co-MOF. Moreover, it further facilitates the separation of photo-excited charge carriers. Moreover, Li et al. have fabricated Co-MOF via the solvothermal method for photocatalytic evolution of H_2_ under irradiation of visible light [39]. The experiment results revealed that the photocatalytic H_2_ evolution rate of the synthesized material reaches a high value of 1.44 mmolg^−1^ h^−1^ in H_2_O solution with Na_2_S and Na_2_SO_3_ as a sacrificial agent without co-catalyst.

### 3.2. Photocatalytic Degradation of Organic Pollutants

Chen et al. have used hybrid modification strategies to construct Co/Ni-MOFs@BiOI catalyst based using a simple hydrothermal synthesis method [117]. In this study, different photoactive semiconductors were also coupled with MOFs constructing multi-heterojunctions for more efficient charge separation. The authors found that the combination of Co/Ni-MOFs with BiOI has significantly improved the capability of absorbing visible light which results in enhancing photocatalytic degradation of methylene blue in aqueous solutions without any chemical additives. They reported that the 20 mg/L of Co/Ni-MOFs@BiOI-8.5% exhibited the highest methylene blue removal of 81.3% within 240 min under irradiation of visible light and showed almost four times higher photocatalytic activity than that of Co/Ni-MOFs. This could be attributed to the absorption wavelength to visible light, the acceleration of charge-carrier separation efficiency as well as a synergetic effect.

In another study, Govindaraju and co-workers successfully used Zn-MOF decorated bio-activated carbon for photocatalytic degradation of methyl orange (86.4%) and brilliant green (78.8%) within 90 min under irradiation of UV light [118]. Hussain et al. showed the highest photocatalytic degradation activity with 99.7% methylene blue removal in 3 h under visible light using N-O-TiO_2_/C composite derived from NH_2_-MIL-125(Ti) [119]. The high photocatalytic performance may be due to the presence of carboxylate functional groups (-COOH) on the surface of the carbon matrix in the composite because these functional groups increase the hydrophilicity of the composite and provide additional active sites which facilitate better accommodation of methylene blue molecules. It may also have attributed to the optimal anatase/rutile phase junctions together with the formation of photo-active oxygen rich N-O like interstitial/intraband states above the valence band of TiO_2_.

Zhang et al. have successfully prepared magnetically recoverable Z-scheme ZnFe_2_O_4_/Fe_2_O_3_ perforated nanotube using a one-step MOF-derived calcination method by using MIL-88B/Zn core/shell nanorod as a precursor [99]. The authors reported that the MIL-88B core reacted with O_2_ to form hollow Fe_2_O_3_ and the Zn^2+^ shell underwent an in situ solid-state reaction with Fe_2_O_3_ to cause volume shrinkage and formation of ZnFe_2_O_4_ and finally turned into ZnFe_2_O_4_/Fe_2_O_3_ perforated nanotube in the calcination process. They found that ZFF-2 exhibited the highest photocatalytic ciprofloxacin degradation performance with a degradation percentage of 96.5% and a TOC removal percentage up to 89% under light irradiation for 180 min. The synthesized material showed no significant decrease in photocatalytic degradation of ciprofloxacin even after five successive recycling experiments. The enhanced photocatalytic performance may be due to the perforated nanotube and Z-scheme transfer pathway, high light absorption ability, and high interfacial separation of photo-generated charge.

In a different study, Hou et al. showed outstanding photocatalytic degradation of perfluorooctanoic acid up to 89.6% within 3 h under optimal conditions using lignin/PVA/bimetallic MOFs composite [108]. Perfluorooctanoic acid is used as an industrial surfactant and as a material feedstock. In addition, the lignin/PVA/bimetallic MOFs composite retained 77% catalytic capacity after 4 cycles.

### 3.3. Photocatalytic Reduction of CO_2_

Photocatalytic reduction of CO_2_ to value added products, such as methane, ethanol, methanol, CO, methanoic acid, etc., has been widely reported. MOFs has recently surfaced as a better candidate for CO_2_ reduction. This is owing to the presence of organic linkers and metal center nodes in MOFs that enable easy absorption of light energy and desirable photoactive catalytic properties for CO_2_ reduction. For instance, Idris and co-authors have demonstrated the photocatalytic reduction of CO_2_ using dye-sensitized 2D Fe-MOF nanosheets [40]. The dye//Fe-MOF nanosheets catalytic system exhibited a good photocatalytic CO_2_-to-CO activity of 1120 μmol g^−1^ h^−1^ under visible light irradiation. Moreover, the photocatalytic CO production was further enhanced by regulating the electronic structure of the 2D Fe-MOF nanosheets by doping with Co ions, achieving a remarkable photocatalytic activity of 1637 μmolg^−1^ h^−1^. In a different study, Xu and co-workers have employed CdS/Ni-MOF for selective photocatalytic reduction of CO_2_ to CO. ^130^Among the prepared materials, 20%-CdS/Ni-MOF showed the best photocatalytic reduction performance, the yield of CO reached 7.47 μmol g in the 4th hour, which was nearly 16 times and 7 times that of Ni-MOF and CdS, respectively. In another study, Sonowal et al. have reported the reduction of CO_2_ to methanol using g-C_3_N_4_@Zr-based MOF under visible light irradiation [120]. The maximum yield of methanol obtained is 386 µmol g^−1^ h^−1^.

In summary, as a novel organic-inorganic hybrid structure, MOFs can function as photocatalysts because they can combine photo-sensitizer and catalytic capabilities in one structure. Photo-active MOFs can function as improved photocatalysts by minimizing contamination, and easily be recovered and reused due to their solid nature. Various factors including tunable energy levels, metal-to-ligand or ligand-to-metal charge transfer transitions effect, and orientation of organic linkers should be further studied and optimized in order to develop stable and high functionality structure and components of the MOF photocatalysts.

## 4. Photocatalytic Mechanisms of MOFs and MOFs-Based Heterostructures

Light absorption in MOFs and MOFs-based heterostructures can occur either through the inorganic metal atom or through the organic linker [121]. MOFs often contain few absorption bands in the UV-Vis region due to the chemical nature of the chromophore centers originate from the inorganic metal atom or organic linkers [122]. In addition, distinct π–π * transitions of the aromatic units in the organic linker or to metal-to-ligand or ligand-to-metal charge transfer transitions often contribute the absorption bands [123,124]. As a result, the semiconductors band theory is not appropriate to describe the light absorption and subsequent transitions observed in MOF materials.

### 4.1. Photocatalysis of Pristine MOFs Structures

According to the principle of the traditional photocatalytic process of semiconductors, an incident light with an energy greater than the band gap energy of a semiconductor can directly excite a photocatalyst. Upon light illumination, charge transfer from ligand (organic site) to metal (inorganic site) occurs in the MOF [125]. Figure 3 illustrated the general photocatalytic mechanism of the MOFs. In the pristine MOFs, e^−^ are excited from the highest occupied molecular orbital (HOMO) to the lowest unoccupied molecular orbital (LUMO) leaving behind h^+^ in the HOMO [126,127]. The HOMO/LUMO functions similarly to the VB/CB in semiconductors in this context [128]. In the LUMO (i.e., CB in a typical semiconductor), the photo-generated e^−^ can be transferred to O_2_, resulting in the generation of ^•^O_2_^−^ radicals. Meanwhile, the surface hydroxyl group/water can be oxidized by h^+^ in the HOMO (i.e., VB in a typical semiconductor), resulting in the formation of ^•^OH. The presence of these reactive species can then be used to degrade different types of pollutants [97]. In a review reported by Ikreedeegh et al., the authors have explained in-depth the redox reactions of the MOF [129]. In brief, the reduction and oxidation of acceptors (CO_2_) and donor species (H_2_O) by the e^−^/h^+^ pair, respectively. However, for MOF semiconductors with small band gap energy, generally, they generate a reverse process and recombine immediately to release unproductive energy in the form of heat.

For instance, Sun et al. have successfully oxidized amines to imines using NH_2_-MIL-125(Ti) under exposure to visible light [130]. The photo-generated Ti^3+^ and ^•^O_2_^−^ which were formed via the reaction between Ti^3+^ and O_2_ were reported to be involved in the transformation process based on the experimental observations and their electron spin resonance result. Chen et al. reported an enhanced photocatalytic CO_2_ reduction performance using NH_2_-MIL-125(Ti) [131]. This is due to the enhanced optical absorption and CO_2_ uptake ability with the introduction of NH_2_BDC. However, it was further mentioned that after the introduction of ZnTCPP, it further enhanced the CO_2_ uptake ability and, hence, significantly increased the photoelectric conversion ability (Figure 4).

### 4.2. Photocatalysis of MOFs-Based Heterostructures

Researchers have used various methods to fabricate visible light responsive MOFs including altering metal nodes, functionalizing organic linkers, employing hybrid linkers, and developing composite photocatalysts in order to maximize visible light harvesting in the solar spectrum. Figure 5 shows a general photocatalytic mechanism of MOFs-based heterostructures. For example, Qiu and co-workers have reported the remarkable photocatalytic degradation of an oral antibiotic, sulfamethoxazole using Ag@Zr-based MOFs [115]. They also reported the low leaching amount of Ag and Zr under different pH values which indicated the excellent stability of the synthesized materials. Under exposure to light, the e^−^ would be enriched on the surface of Ag nanoparticles (Figure 5). Subsequently, the photo-excited e^−^ would transfer from Ag nanoparticles to Zr-based MOFs to further react with O_2_ to produce ^•^O_2_^−^. On the other hand, the photo-excited h^+^ could react with H_2_O to generate O_2_. In addition, some generated ^•^O_2_^−^ could be further transferred to react with ^•^OH radicals. All of the oxygen-active species took part in the degradation of sulfamethoxazole.

In a different study, Li et al. fabricated 2D porphyrin Co-TCPP MOF decorated 2D B-TiO_2−X_ nanosheet (Co-TCPP MOF@B-TiO_2−X_) photocatalysts via the hydrothermal method [114]. The prepared photocatalysts exhibited outstanding photocatalytic performance for the degradation of bisphenol A up to 97% within 120 min irradiation. The degradation rate of the composite was 7.3 and 19.3 times higher than Co-TCPP MOF and B-TiO_2−X_, respectively. They reported that both B-TiO_2−X_ and Co-TCPP MOF could concurrently produce photogenerated e^−^/h^+^ pairs upon irradiation of visible light. Based on the conventional type II heterojunction, the photogenerated e^−^ are transferred from the LUMO of Co-TCPP MOF to the CB of B-TiO_2−X_ due to the more negative LUMO position of Co-TCPP MOF. Meanwhile, due to the more positive VB position of B-TiO_2−X_, the photogenerated h^+^ were transferred from the VB of B-TiO_2−X_ to the HOMO of Co-TCPP MOF indicating that these heterostructures are effective in suppressing the recombination of photogenerated carriers and boosting the photocatalytic performance.

However, according to a recent literature review by Behera et al., there are four types of possible mechanisms by various MOFs based photocatalysts: (i) Heterojunction based charge dynamics, (ii) Z-scheme based charge dynamics, (iii) Cocatalyst mediated charge dynamics, and (iv) Elemental doping based charge dynamics [132]. For instance, MOF-derived ZnFe_2_O_4_/Fe_2_O_3_ nanotube showed a direct Z-scheme rather than a typical heterojunction photocatalyst [99]. Based on the optical and band properties determined by DRS and Mott-Schottky, as well as fluorescence analysis, a corresponding photocatalytic CIP degradation mechanism has been proposed. The photogenerated e^−^ on the CB of Fe_2_O_3_ will pass through the Fe_2_O_3_/ZnFe_2_O_4_ interface which, therefore, will recombine with the photogenerated holes on the VB of ZnFe_2_O_4_. The heterojunction facilitates the separation of photogenerated e^−^/h^+^ pairs and maintains the strong oxidizing holes on the VB of Fe_2_O_3_ and highly reductive electrons on the CB of ZnFe_2_O_4_. Therefore, in the presence of visible light and ZnFe_2_O_4_/Fe_2_O_3_ composites, ^•^O_2_^−^ and ^•^OH are thermodynamically generated in the photocatalytic CIP degradation system (Figure 6). Other studies that proposed the Z-scheme have been reported by Qiao et al. for hydrogen evolution from photocatalytic water splitting in the MOF/MoS_2_ photocatalyst and by Wang et al. for photocatalytic removal of alkylphenol ethoxylate using CuO@Cu-based MOFs composite [101,133]. Photocatalytic reduction of Cr(VI) using NiFe_2_O_4_/MOF-808 composite was reported by Khosroshahi et al. [134]. The mechanism of photocatalytic reduction of Cr(VI) has been proposed using the Z-scheme pathway as described earlier.

On the other hand, Cao et al. reported an S-scheme heterojunction for photocatalytic hydrogen evolution using Cu-MOFs modifies Mn_0.05_Cd_0.95_S NPs [135]. The charge transfer mode has been explained using Z-scheme and S-scheme (Figure 7) [136]. For Z-scheme, semiconductor II (SC II) with a higher Fermi level to ensure its electron flow to semiconductor I (SC I) through the interface of SC II-SC I until both Fermi levels of SC I and SC II are eventually at the same level [136]. In addition, the positive and negative charges gather at interfaces near SC II and SC I, respectively, which produces an internal electric field (IEF). Then, from CB of SC I, photogenerated e^−^ are transferred to SC II VB with IEF action. For the S-scheme principle, the difference in Fermi level increases of SC II band edge bending for e^−^ supply layer formation. This leads to the decrease of the SC I band edge for electron acquisition. The difference in Fermi levels between SC I and SC II results in the transfer of the photogenerated e^−^ from CB of SC I to VB of SC II through the bending charge transfer channel within IEF. Therefore, in this case, Cao et al. mentioned that under visible light irradiation, the photogenerated e^−^ from the CB of the Cu-MOFs and holes from the VB of the MCS are recombined under the internal electric field, band edge bending, and coulomb interaction. The S-scheme pathway shows strong redox capacity, thus providing the powerful driving force for photocatalytic H_2_ evolution [135].

In summary, MOFs and MOFs-based heterostructures can absorb light either through the inorganic metal atom or through the organic linker. Under light exposure, e^−^ are photoexcited from the highest occupied molecular orbital (HOMO) to the lowest unoccupied molecular orbital (LUMO) leaving behind h^+^ in the HOMO in pure MOFs materials. However, to maximize visible light harvesting in the solar spectrum, the properties of MOFs have been altered by altering metal nodes, functionalizing organic linkers, employing hybrid linkers, and developing composite photocatalysts.

## 5. Future Outlook

To summarize, MOFs are emerging as the new promising materials in the field of photocatalysis. However, they still have some limitations. The following future recommendations should be considered to address all these issues and challenges:Firstly, most MOF synthesis methods require high production costs which may restrict their applications in various fields. Therefore, a simple and cost-effective synthesis method to prepare MOFs should be introduced;Recyclability, reusability, and stability of MOFs have not been discussed widely. These qualities are crucial for preparing MOFs with excellent properties;In-depth study of the physicochemical properties of these materials, such as light absorption in the visible region, electronic structure, crystallographic properties, porosity, and surface area, is highly required with the aid of advanced characterization techniques;MOFs usually exist in powder form which leads to poor stability and reusability. Recycling issue of MOFs should be addressed properly. For example, MOFs fixed to wood channels can solve the recycling problems as stated by Ma et al. [137,138];More detailed studies to improve the stability of MOFs by in-situ growth of MOFs on wood should be carried out;Huge efforts should be exerted by carrying out more intensive research works on the determination of the MOFs photocatalytic mechanisms such as the Z-schemes and S-schemes since it is considered to be the key to developing and engineering photocatalysts with efficient charge separation and transport.

## 6. Conclusions

MOFs have emerged as novel photocatalysts owing to their structural characteristics of large surface area, abundant active sites as well as well-ordered porous structure. However, fast recombination of electron-hole pairs has become one of the main challenges to provide an effective photocatalyst. Therefore, in this review, an overview of synthesis methods used in the fabrication of MOFs and MOFs-based heterostructures for various potential photocatalytic applications were comprehensively compiled and discussed. Mechanisms of photocatalytic performance using MOFs and MOFs-based heterostructures, such as generation of the charge carriers, separation of e^−^/h^+^ pairs, and the degradation/removal pathways of representative pollutants have been illustrated in detail. However, an improvement in the overall characteristics of MOFs and MOFs-based heterostructures as visible light active materials should be the focus in a future study. Hence, research gaps regarding the development of MOFs and MOFs-based heterostructures were highlighted in the future prospects at the end of this review.

## Figures and Tables

**Figure 1 nanomaterials-12-02820-f001:**
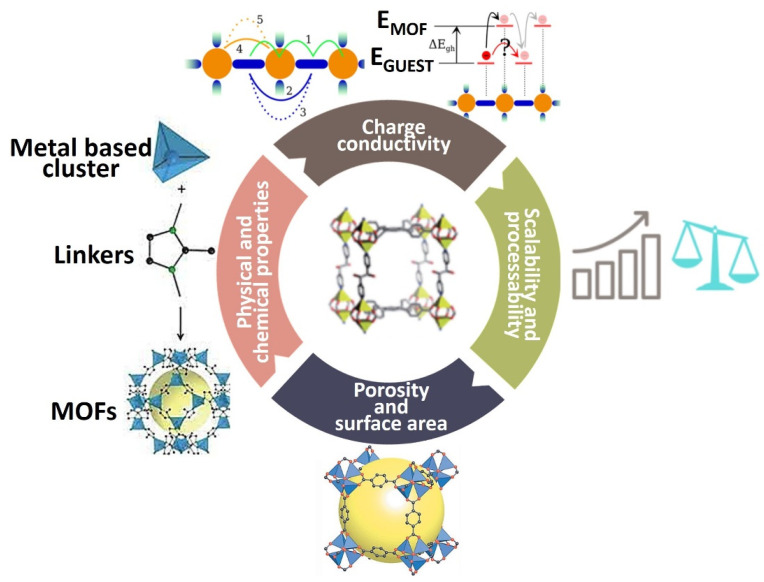
Tunable properties of MOF during a synthesis.

**Figure 2 nanomaterials-12-02820-f002:**
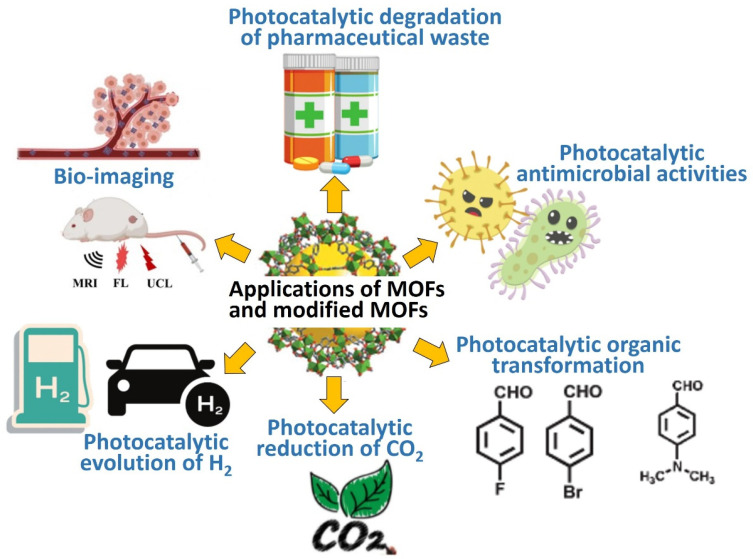
Different applications of MOFs and MOFs-based heterostructures.

**Figure 3 nanomaterials-12-02820-f003:**
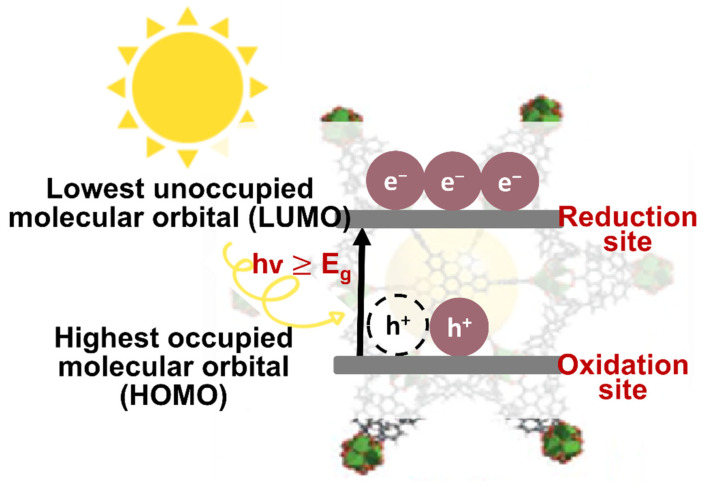
The mechanism of photocatalysis of pristine MOFs.

**Figure 4 nanomaterials-12-02820-f004:**
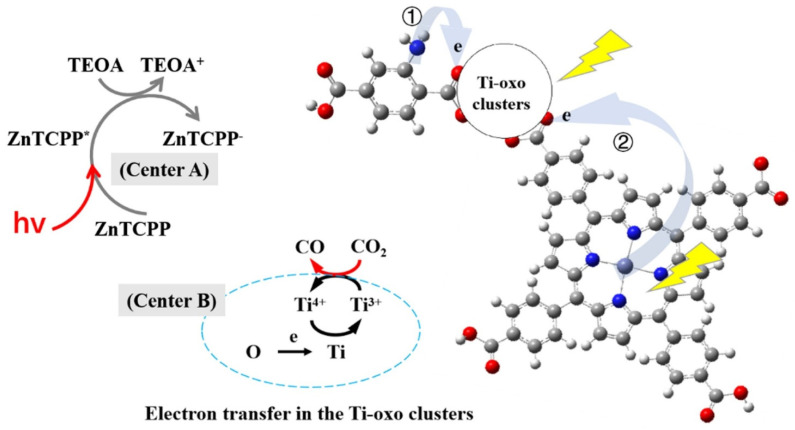
The reaction mechanism of photocatalytic CO_2_ reduction using D-TiMOF. Reprinted with permission from Ref. [131]. Copyright 2022 Elsevier.

**Figure 5 nanomaterials-12-02820-f005:**
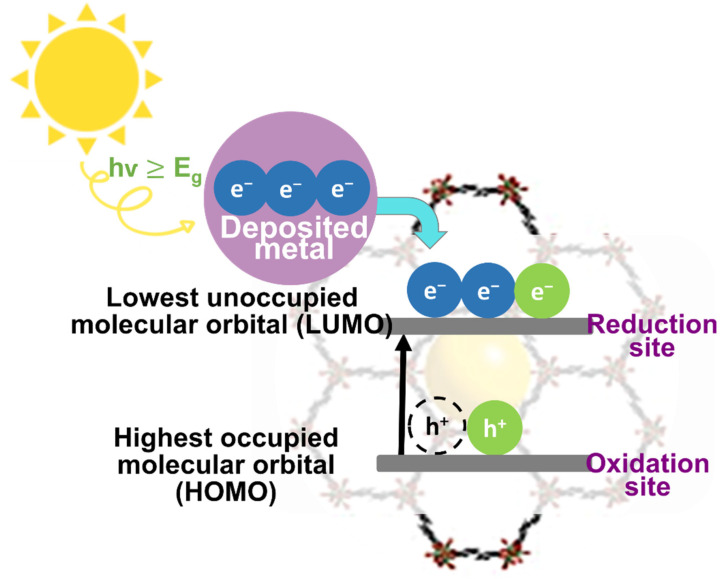
Example of photocatalysis mechanism of MOFs-based heterostructures.

**Figure 6 nanomaterials-12-02820-f006:**
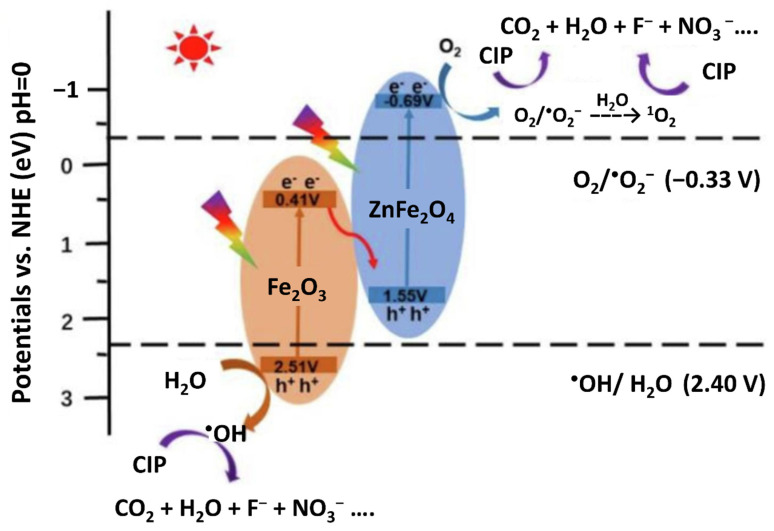
Z-scheme of photocatalytic CIP degradation system. Reprinted with permission from Ref. [99]. Copyright 2022 Elsevier.

**Figure 7 nanomaterials-12-02820-f007:**
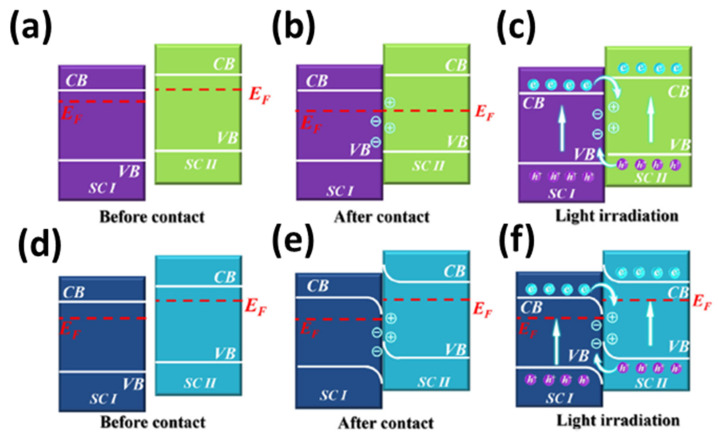
(**a**–**c**) Z-scheme and (**d**–**f**) S-scheme photocatalytic systems. Reprinted with permission from Ref. [136]. Copyright 2022 John Wiley and Sons.

**Table 1 nanomaterials-12-02820-t001:** Synthesis of MOF materials using various methods.

No.	Materials	Synthesis Methods	Average Particle Size	Ref.
1	PEGylated iron trimesate (materials institute Lavoisier) MIL-100 (Fe)	Hydrothermal	129–174 nm	[52]
2	Rare-earth 3D cluster-based MOFs	Hydrothermal	0.4–2.7 μm	[54]
3	2D-MOFs with Ni(II), Cu(II), and Co(II)	Hydrothermal	-	[55]
4	Zn_4_O(BDS)_3_ MOFs	Solvothermal	807 nm	[49]
5	Cu(II)-based MOFs	Solvothermal	50 nm	[57]
6	Zr-based MOFs	Microwave	Less than 100 nm	[51]
7	Cu-BTC MOFs	Room temperature template method	-	[50]
8	UiO-66 MOFs	Vapor-assisted	200 nm	[58]
9	Copper dicarboxylate MOFs	Chemical vapor deposition	-	[59]
10	Zn-based MOFs	Sonochemical	-	[61]

**Table 2 nanomaterials-12-02820-t002:** Different modifications of MOFs.

No.	MOFs-Based Heterostructures	Synthesis Method	Band Gap Energy	Photocatalytic Activities	Photocatalytic Performance	Ref.
1	BiPO_4_/Fe-metal organic framework composite	Solvothermal method	Pristine NH_2_-MIL-53(Fe): 2.11 eV BiPO_4_: 3.75 eV	Photocatalytic degradation of tetracycline hydrochloride, Indigo Carmine and reduction of 4-nitrophenol	Tetracycline hydrochloride: degraded 80% within 120 min Indigo Carmine: degraded 94% within 120 min 4-nitrophenol: 95% reduction efficiency within 12 min	[66]
2	CdSe QDs@ Fe-based metal-organic framework composites	Heated to reflux in oil bath	CdSe: 1.97 eV Fe-BDC: 2.71 eV	Photocatalytic degradation of rhodamine B	Almost completely degraded ~99.8% after 240 min	[75]
3	Copper-mediated metal-organic framework	Advanced double-solvent approach followed by one-step reduction	-	Photocatalytic oxidation of aromatic alcohols	Up to 69.4% conversion	[76]
4	Graphitic carbon nitride nanosheet/metal-organic framework	Hydrothermal	g-C_3_N_4_: 2.72 eV MIL-101(Fe): 2.55 eV g-C_3_N_4_/MIL-101(Fe): 2.05 eV	Photocatalytic degradation of rhodamine B	MIL-101(Fe)/g-C_3_N_4_: 99.3% Bare g-C_3_N_4_: 40% Bare MIL-101(Fe): 73%	[77]
5	CdS in a Ti, Zr-based metal-organic framework	Stepwise precipitation	P415: 3.47 eV P415-NH_2_: 2.30 eV CdS: 2.41 eV	Photocatalytic production of H_2_	The rate of H_2_ production: 5.8 μmol min^−1^g^−1^	[78]
6	Hydrogels encapsulating Gold/metal-organic frameworks nanocomposites	Schiff base reaction and radical polymerization	ZIF-8: 3.19 eV	Photocatalytic antibacterial and wound healing activities	Au@ZIF-8@GCOA: up to 99.1% for *E. coli* and 99.6% for *S. aureus* after 20 min of irradiation	[35]
7	Modified Ag_3_VO_4_ with metal-organic frameworks	Facile two-step method	Ag_3_VO_4_: 2.0 eV 20%AZ: 2.11 eV 40%AZ: 2.21 eV 60%AZ: 2.25 eV 80%AZ: 2.83 eV ZIF-8: 5.06 eV	Photocatalytic degradation of rhodamine B	60%AZ composites exhibited the highest, about 4.2 times of pure Ag_3_VO and 22.8 times of bare ZIF-8, respectively	[79]
8	Metal-organic framework g-C_3_N_4_/MIL-53(Fe) heterojunctions	Solvothermal method	MIL-53(Fe): 2.72 eV CMFe-3: 2.51 eV	Photocatalytic reduction of Cr(VI)	g-C_3_N_4_/MIL-53(Fe) showed about 2.1 and 2.0 times higher photocatalytic efficiency for the reduction of Cr(VI) in comparison to pure g-C_3_N_4_ and MIL-53(Fe), respectively.	[80]
9	Metal-organic frameworks modified with Bi_2_WO_6_ nanosheet	Hydrothermal method	-	Photocatalytic degradation of tetracycline	12%MIL/BWO achieved the highest removal efficiency of about 92.4% within 120 min	[81]
10	Titanium dioxide/magnetic metal-organic framework	Hydrothermal method	TiO_2_: 3.1 eV TiO_2_/magnetic MIL-101(Cr): 1.61 eV	Photocatalytic degradation of acid red 1	TiO_2_/magnetic MIL-101(Cr) showed 90% degradation of acid red 1	[82]
11	Modified stannous sulfide nanoparticles with a metal-organic framework	Deposition method at room temperature	pure MIL-53(Fe): 1.6 eV SnS: 0.87 eV	Photocatalytic reduction of chromium (VI)	71.3% of Cr(VI) removal is achieved after 60 min	[83]
12	UiO-66-based metal-organic frameworks	Solvothermal method	UiO-66: 3.82 eV UiO-66-NH_2_: 2.63 eV UiO-66-(OH)_2_: 2.50 eV	Photocatalytic degradation of acetaminophen	90% after 6 h	[84]
13	Lanthanide-organic-frameworks modified TiO_2_	Solvothermal method	TiO_2_: ~3.2 eV Ln(ndc) MOF: ~2.9 eV	Photocatalytic degradation of phenol	87.5% after 60 min	[85]
14	Fe/Ce-based bimetallic MOF	Dielectric barrier discharge plasma	Ce-MOF: 3.01 eV Fe/Ce-MOF-1: 1.90 eV Fe/Ce-MOF-2: 1.97 eV Fe/Ce-MOF-3: 1.75 eV	Photocatalytic degradation of methyl orange	Fe/Ce-MOF-2 could degrade 93% methyl orange in 30 min under visible light	[86]

**Table 3 nanomaterials-12-02820-t003:** Different photocatalytic applications of MOFs and MOFs-based heterostructures.

No.	Synthesized Materials	Band Gap Energy	Applications	Source of Light	Efficiency	Ref.
1	Sm doped 2D Co-MOF/ protonated-g-C_3_N_4_	g-C_3_N_4_: 2.82 eV Protonated-g-C_3_N_4_: 2.70 eV 2D Co-MOF: 3.12 eV	Photocatalytic evolution of hydrogen	300 W Xe lamp with an AM 1.5G filter	The H_2_ evolution rate of 5% Sm doped with the highest activity has reached 73.42 μmolh^−1^.	[98]
2	MOF-derived magnetically recoverable ZnFe_2_O_4_/Fe_2_O_3_ perforated nanotube	Fe_2_O_3_: 2.10 eV ZnFe_2_O_4_: 2.24 eV	Photocatalytic removal of ciprofloxacin	300 W Xe lamp with AM 1.5 filter	ZFF-2 exhibits the best photocatalytic ciprofloxacin degradation performance with a degradation percentage of 96.5% and a TOC removal percentage up to 89% under light irradiation for 180 min	[99]
3	MOF derived carbon modified porous TiO_2_	C-TiO_2_-3: 2.45 eV C-TiO_2_-4: 2.85 eV C-TiO_2_-5: 2.94 eV C-TiO_2_-6: 2.37 eV C-TiO_2_-7: 2.93 eV	Photocatalytic oxidation of cyclohexane	100 W mercury lamp with a UV cutoff filter (λ ≥ 420 nm)	C-TiO_2_-4 catalyst, the yield of cyclohexanol and cyclohexanone could reach 13.18 and 107.07 μmol respectively, and the selectivity to cyclohexanone reaches 89.0%.	[100]
4	Polymer-tethered Zr-porphyrin MOFs encapsulated carbon dot nanohybrids	PCN-222: 1.75 eV Citric acid derived QDs@PCN-222: 1.64 eV Nitrogen-doped CDs@PCN-222: 1.59 eV Sulfur- doped CDs@PCN-222: 1.66 eV	Photocatalytic degradation of rhodamine B and tetracycline	300 W UV lamp with a 420 filter	The removal efficiency of rhodamine B and tetracycline reached almost 100% and 90.93%, respectively under 20 min visible light irradiation	[101]
5	Marigold-like CuO@Cu-based MOFs composite	CuO: 1.41 eV Cu-H_3_BTC MOF: 2.0 eV	Photocatalytic removal of alkylphenol ethoxylate	300 W Xe lamp with a 420 nm cut-off filter	CuO@Cu-H_3_BTC MOF composite exhibited up to 79% removal of alkylphenol polyethoxylate within 120 min of visible light irradiation	[102]
6	Citrate capped Fe_3_O_4_@ UiO-66-NH_2_ MOF	Citrate capped Fe_3_O_4_: 1.45 eV UiO-66-NH_2_: 2.67 eV	Adsorption of Cr (VI) and photocatalytic evolution of H_2_	300 W Xe lamp	MU-2 showed a maximum monolayer adsorption capacity of 743 mg g^−1^ which followed pseudo-second-order kinetics. In addition, the synthesized composite material displayed enhanced activity towards photocatalytic H_2_ evolution with a maximum evolution rate of 417 μmolh^−1^ with an apparent conversion efficiency of 3.12%.	[103]
7	MOF/ reduced graphene oxide hybrid	Not calculated	Photocatalytic degradation of methylene blue, rhodamine B and methyl orange	Solar simulator with wavelengths in the range of 300–1900 nm	MOF-5@rGO photocatalytic degradation efficiency reached 93% after 20 min illumination	[104]
8	ZIF-8, UiO-66 and Cu-BTC are adopted to combine with rGO and TiO_2_	Uio-66-rGO/TiO_2_: 2.6 eV Cu-BTC-rGO/TiO_2_: 2.80 eV ZIF-8-rGO/TiO_2_: 2.90 eV	Photocatalytic degradation of rhodamine B	500 W Xe lamp and a cutoff filter (λ > 400 nm)	Degradation rate constants of rhodamine B for UiO-66-RGO/TiO_2_ reached 1.65 × 10^−1^min^−1^ and 1.12 × 10^−1^ min^−1^ under UV- and visible-light irradiation, respectively.	[105]
9	Cd-MOFs	Not calculated	Photocatalytic evolution of H_2_	300 W Xe lamp	The highest H_2_ production rate reached 17,242 μmolg^−1^ h^−1^	[106]
10	Co/Ni-MOFs@ BiOI composite	Not calculated	Photocatalytic degradation of crystal violet, rhodamine 6G, malachite green, Congo red and methyl orange	300 W PLS- SXE with a cut-off filter (λ = 420 nm)	The removal efficiency of Co/Ni-MOFs@BiOI-8.5% followed the order of MG > MB > MO > CR > CV > R6G, and the degradation rate constants (k, min^−1^) were 0.04705, 0.00627, 0.00681, 0.00695, 0.00347, 0.00066, respectively.	[91]
11	MIL-101(Fe) derived Fe_2_O_3_ with TiO_2_	Fe_2_O_3_: 1.7–2.0 eV TiO_2_: 3.17 eV FeTi-3%: 2.94 eV	Photocatalytic degradation of a mixture of nonsteroidal anti-inflammatory drugs namely ibuprofen and naproxen	Philips 25 W/m^2^, λ = 365–700 nm	The material exhibited up to 91% and 100% degradation of ibuprofen and naproxen within 240 and 15 min of reaction, respectively.	[107]
12	Ti-MOF/ plasmonic Ag nanoparticle/ NiFe layered double hydroxide	Ti-MOF: 2.53 eV NiFeLDH: 2.19 eV Ti-MOF/ NiFeLDH: 2.41 eV Ti-MOF/Ag/ NiFeLDH-1: 2.17 eV Ti-MOF/Ag/ NiFeLDH-2: 2.09 eV Ti-MOF/Ag/ NiFeLDH-3: 2.12 eV Ti-MOF/Ag/ NiFeLDH-4: 2.15 eV	Photocatalytic degradation of antibiotics, levofloxacin and rhodamine B dye	300 W Xe With a cut-off filter of λ > 420 nm	Ti-MOF/Ag/NiFeLDH composite displayed outstanding photocatalytic degradation up to 95% for rhodamine B removal within 50 min and 92% levofloxacin degradation efficiency in 70 min	[34]
13	Dye sensitized Fe-MOF nanosheets	Not calculated	Photocatalytic reduction of CO_2_	300 W Xe lamp with a 420 nm cut-off filter	The synthesized material exhibited a significant photocatalytic CO production rate of 1120 μmol g^−1^ h^−1^	[40]
14	Lignin-based bimetallic MOFs nanofibers composite membranes with peroxymonosulfate	2.66 eV	Photocatalytic degradation of perfluorooctanoic acid	9 W UV lamp with the wavelength of photoexcitation is 185 nm and under solar light	Lignin/PVA/bimetallic-MOFs showed outstanding performance up to 89.6% degradation of perfluorooctanoic acid within 3 h	[108]
15	Pt photo deposited on MIL-53(Fe)	Not calculated	Photocatalytic evolution of H_2_	300 W Xe lamp	Deposition of Pt nanoparticles on Fe-MOFs can lower the overpotential for H_2_ evolution toward further enhanced photocatalytic activity.	[109]
16	BiOIO_3_/ MIL-88B	BiOIO_3_: 2.88 eV MIL-88B: 2.31 eV BMIL-5: 2.48 eV	Photocatalytic degradation of Reactive Blue 19 and tetracycline hydrochloride	300 W Xe lamp	The BiOIO_3_/MIL-88B composites exhibited an excellent removal rate for Reactive Blue 19 and tetracycline hydrochloride under visible light irradiation, which was approximately 3.28 and 4.22 times higher than the pristine BiOIO_3_, respectively.	[110]
17	Porphyrin- Zr MOF	Not calculated	Photocatalytic degradation of bisphenol F	500 W Xe lamp with a 420 nm cut off filter	The material could achieve over 78% BPF removal (with/without salt) after 8 cycles.	[83]
18	Graphitic carbon nitride/NH_2_-MIL-101(Fe)	Fe-MOF: 1.64 eV g-C_3_N_4_: 2.77 eV Composite: 1.90 eV	Photocatalytic degradation of acetaminophen and reduction of Cr(VI)	Solar light (60,000 lux)	The composite showed the highest degradation of acetaminophen 94% at pH 7 and Cr(VI) reduction efficiency of 91% at pH 2	[111]
19	MIL-100 (Fe) MOF/MOX homojunction	Not calculated	Photocatalytic oxidation of gaseous benzene, toluene, xylenes and styrene	250 W Xe lamp	MIL-100(Fe) MOF/MOX homojunction showed up to 23%, 41%, 82%, 79% and 83% photocatalytic oxidation of benzene, toluene, p-xylene, m-xylene and styrene, respectively	[112]
20	CuO-ZnO/ ZiF-8 MOF	CuO-ZnO: 2.89 eV CuO-ZnO/ZiF-8(20): 1.96 eV ZiF-8: 5.34 eV	Photocatalytic degradation of acid orange 7	400 W halogen lamp	CuO-ZnO/ZiF-8(20) showed the highest photocatalytic degradation of 98.1% acid orange 7 in 100 min	[113]
21	Co-TCPP MOF@B- TiO_2−X_	pure B-TiO_2−x_: 2.71 eV Co-TCPP MOF: 2.55 eV BTC-10%: 2.24 eV	Photocatalytic degradation of bisphenol A	A 300W Xe lamp with a 420nm cut- off filter	Co-TCPP MOF@B-TiO_2−X_ exhibited remarkable photocatalytic degradation of bisphenol A up to 97% within 120 min irradiation	[114]
22	Ag@Tetra thiafulvalene-based Zr-MOF	Not calculated	Photocatalytic degradation of sulfamethoxazole	500 W Xe lamp	Ag NPs@Zr-TTFTB were found to efficiently remove sulfamethoxazole (k = 0.009 min^−1^)	[115]
23	CdS/Ni-MOF heterostructure	CdS: 2.0 eV Ni-MOF: 3.3 eV	Photocatalytic reduction of CO_2_	300 W Xe lamp	20%-CdS/Ni-MOF showed the best photocatalytic reduction performance, the yield of CO reached 7.47 μmol/g in the 4th hour, which was nearly 16 times and 7 times that of Ni-MOF and CdS	[116]

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
