# Peer review of "Recent Progress of Metal-Organic Frameworks and Metal-Organic Frameworks-Based Heterostructures as Photocatalysts"

_nanomaterials, 2022, doi:10.3390/nano12162820_

Round 1

Reviewer 1 Report

Here are my comments on the manuscript titled “Recent Progress of Metal-Organic Frameworks and Metal-Organic Frameworks-based heterostructures as Photocatalysts”.

Decision: Major Revision

Mohammad and coworkers aim to provide the overview of recent progress of MOFs and MOFs-based heterostructures on photocatalytic application. The manuscript is easily understandable and does a good job of collecting relevant papers with a focus on state-of-the-art. It is a good fit for nanomaterials. However, there are several aspects that I believe can be improved, as given below in no order.

1.    There are duplications in the keywords. Metal-organic frameworks is same to MOFs. Photocatalysis is identical to photocatalysts.

2.    For Section 2.1 and 2.2, the authors mentioned different methods of synthesis and modification of MOFs, respectively. However, to make it clearly, it is better to use subtitle as follow:

Section 2.1.1 Hydrothermal

Section 2.1.2 Solvothermal

Section 2.1.3 Precipitation

3.    In Page 6, the authors summarized the hydrothermal method to synthesize MOFs. However, in Line 144, “Zr-based metal-organic frameworks were prepared using microwave-assisted synthesis”. It is not proper to cite this reference here. Same to the case listed in Line 148.

4.    References should be cited in Line 5 to Line 8 of Page 13.

5.    As for Section 3, it is suggested to divide into few subsections such as photocatalytic hydrogen generation, CO2 reduction, and organic pollutant degradation.

6.    In Section 5, the authors only mentioned some issues and challenges should be address without giving useful solutions.

7.    Please read thoroughly and carefully revise the manuscript.

8.    Few terms appear first without detailed explanation such as RSO and LUCO.

9.  As for references, lots of errors should be corrected including missing volume, issue, and page numbers, subscripts of term, and improper abbreviations. It is recommended to downloading these citations from the official website instead of google scholar.

Author Response

Reviewer 1:

Comments and Suggestions for Authors

Here are my comments on the manuscript titled “Recent Progress of Metal-Organic Frameworks and Metal-Organic Frameworks-based heterostructures as Photocatalysts”.

Decision: Major Revision

Mohammad and coworkers aim to provide the overview of recent progress of MOFs and MOFs-based heterostructures on photocatalytic application. The manuscript is easily understandable and does a good job of collecting relevant papers with a focus on state-of-the-art. It is a good fit for nanomaterials. However, there are several aspects that I believe can be improved, as given below in no order. 

Reply: Thank you for your nice comments. We appreciate your suggestions and recommendations for the publication of this manuscript.

  1. There are duplications in the keywords. Metal-organic frameworks is same to MOFs. Photocatalysis is identical to photocatalysts.

Reply: Thank you for pointing this out. We have removed MOFs in the keywords.

  1. For Section 2.1 and 2.2, the authors mentioned different methods of synthesis and modification of MOFs, respectively. However, to make it clearly, it is better to use subtitle as follow:

Section 2.1.1 Hydrothermal

Section 2.1.2 Solvothermal

Section 2.1.3 Precipitation

Reply: Thank you for your suggestion. We have organized and categorized the synthesis methods with different subheadings on pages 5-8.

  1. In Page 6, the authors summarized the hydrothermal method to synthesize MOFs. However, in Line 144, “Zr-based metal-organic frameworks were prepared using microwave-assisted synthesis”. It is not proper to cite this reference here. Same to the case listed in Line 148.

Reply: Thank you for pointing this out. We have changed it to appropriate sub-headings as suggested. The microwave-assisted synthesis part has been categorized in section 2.1.1 on page 5.

  1. References should be cited in Line 5 to Line 8 of Page 13.

Reply: Thank you for pointing this out. We have added references on line 5 to line 8. The references are as follows:

(98)     Somnath; Ahmad, M.; Siddiqui, K. A. Synthesis of Mixed Ligand 3D Cobalt MOF: Smart Responsiveness towards Photocatalytic Dye Degradation in Environmental Contaminants. Journal of Molecular Structure 2022, 1265, 133399. https://doi.org/10.1016/j.molstruc.2022.133399.

(99)     Shan, C.; Zhang, X.; Ma, S.; Xia, X.; Shi, Y.; Yang, J. Preparation and Application of Bimetallic Mixed Ligand MOF Photocatalytic Materials. Colloids and Surfaces A: Physicochemical and Engineering Aspects 2022, 636, 128108. https://doi.org/10.1016/j.colsurfa.2021.128108.

(100)   Li, M.; Yuan, J.; Wang, G.; Yang, L.; Shao, J.; Li, H.; Lu, J. One-Step Construction of Ti-In Bimetallic MOFs to Improve Synergistic Effect of Adsorption and Photocatalytic Degradation of Bisphenol A. Separation and Purification Technology 2022, 298, 121658. https://doi.org/10.1016/j.seppur.2022.121658.

(101)   Han, W.; Shao, L.-H.; Sun, X.-J.; Liu, Y.-H.; Zhang, F.-M.; Wang, Y.; Dong, P.-Y.; Zhang, G.-L. Constructing Cu Ion Sites in MOF/COF Heterostructure for Noble-Metal-Free Photoredox Catalysis. Applied Catalysis B: Environmental 2022, 317, 121710. https://doi.org/10.1016/j.apcatb.2022.121710.

(102)   Ye, Z.; Feng, S.; Wu, W.; Zhou, Y.; Wang, Y.; Dai, X.; Cao, X. Synthesis of Double MOFs Composite Material for Visible Light Photocatalytic Degradation of Tetracycline. Solid State Sciences 2022, 127, 106842. https://doi.org/10.1016/j.solidstatesciences.2022.106842.

(103)   Lv, S.-W.; Liu, J.-M.; Li, C.-Y.; Zhao, N.; Wang, Z.-H.; Wang, S. Two Novel MOFs@COFs Hybrid-Based Photocatalytic Platforms Coupling with Sulfate Radical-Involved Advanced Oxidation Processes for Enhanced Degradation of Bisphenol A. Chemosphere 2020, 243, 125378. https://doi.org/10.1016/j.chemosphere.2019.125378.

(104)   Guo, J.; Liang, Y.; Liu, L.; Hu, J.; Wang, H.; An, W.; Cui, W. Noble-Metal-Free CdS/Ni-MOF Composites with Highly Efficient Charge Separation for Photocatalytic H2 Evolution. Applied Surface Science 2020, 522, 146356. https://doi.org/10.1016/j.apsusc.2020.146356.

(105)   Liu, X.; Zhang, L.; Li, Y.; Xu, X.; Du, Y.; Jiang, Y.; Lin, K. A Novel Heterostructure Coupling MOF-Derived Fluffy Porous Indium Oxide with g-C3N4 for Enhanced Photocatalytic Activity. Materials Research Bulletin 2021, 133, 111078. https://doi.org/10.1016/j.materresbull.2020.111078.

(106)   Lu, W.; Duan, C.; Liu, C.; Zhang, Y.; Meng, X.; Dai, L.; Wang, W.; Yu, H.; Ni, Y. A Self-Cleaning and Photocatalytic Cellulose-Fiber- Supported “Ag@AgCl@MOF- Cloth’’ Membrane for Complex Wastewater Remediation. Carbohydrate Polymers 2020, 247, 116691. https://doi.org/10.1016/j.carbpol.2020.116691.

(107)   Mahmoud Idris, A.; Jiang, X.; Tan, J.; Cai, Z.; Lou, X.; Wang, J.; Li, Z. Dye-Sensitized Fe-MOF Nanosheets as Visible-Light Driven Photocatalyst for High Efficient Photocatalytic CO2 Reduction. Journal of Colloid and Interface Science 2022, 607, 1180–1188. https://doi.org/10.1016/j.jcis.2021.09.014.

(108)   Wang, Q.; Gao, Q.; Al-Enizi, A. M.; Nafady, A.; Ma, S. Recent Advances in MOF-Based Photocatalysis: Environmental Remediation under Visible Light. Inorganic Chemistry Frontiers 2020, 7 (2), 300–339. https://doi.org/10.1039/C9QI01120J.

  1. As for Section 3, it is suggested to divide into few subsections such as photocatalytic hydrogen generation, CO2reduction, and organic pollutant degradation.

Reply: Thank you for your suggestion. We have divided and reorganized the subsections in the application part on pages 14-21.

  1. In Section 5, the authors only mentioned some issues and challenges should be address without giving useful solutions.

Reply: Thank you for pointing this out. Usually, in the future prospects, we can only suggest other researchers to take into account the current gaps of MOFs in the photocatalysis field as a guide in their research.

  1. Please read thoroughly and carefully revise the manuscript.

Reply: Thank you. We have checked and revised the manuscript accordingly.

  1. Few terms appear first without detailed explanation such as RSO and LUCO.

Reply: Thank you for your pointing it out. We have explained the abbreviated terms on page 9, LUCO as the highest unoccupied crystal orbital, and ROS as reactive oxygen species on line 15 and line 19 (page 9) respectively. However, we have not used abbreviations such as RSO throughout the manuscript.

  1. As for references, lots of errors should be corrected including missing volume, issue, and page numbers, subscripts of term, and improper abbreviations. It is recommended to downloading these citations from the official website instead of google scholar.

Reply: Thank you for your suggestion. We have checked and corrected the references accordingly.

Reviewer 2 Report

Based on the synthesis and application of MOFs and MOFs -based heterostructures photocatalyst, the manuscript comprehensively introduces the synthesis and modification methods of MOFs and MOFs -based heterostructures. The application and mechanism of the materials as photocatalysis are emphasized.  In addition, the manuscript collects and summarizes the literature related to the field comprehensively, and the mechanism of action is well explained through the Mechanism diagrams. However, there are still some issues to be addressed. A moderate revision is suggested before its acceptance.

1.      The introduction of MOFs as photocatalysts in the Introduction section emphasizes the advantages of 3D MOFs, but the structure and mechanism of 2D MOFs seem to be ignored.

2.      More background on the photocatalysis should be provided with more supporting recent articles: Hydrothermal Synthesis of Ce-doped ZnO Heterojunction Supported on Carbon Nanofibers with High Visible Light Photocatalytic Activity; A short review on heterojunction photocatalysts: Carrier transfer behavior and photocatalytic mechanisms; etc. In addition, authors listed many applications of MOFs. One more important application of lithium storage should also be referred: Hierarchical porous Co3O4 nanocages with elaborate microstructures derived from ZIF-67 toward lithium storage.

3.      Topological structure is an important aspect of the spatial structure of MOFs, and at the same time has an important impact on its performance. But this factor has not been mentioned in the article.

4.      In 2.1. Synthesis of MOFs, the introduction of various methods of synthesis is slightly illogical, and it is recommended to follow a certain logical order, such as the synthesis conditions are harsh to simple, etc.

5.      How to construct the MOFs-based heterostructures are important. One more sub-section should be added into the manuscript.

6.      In the manuscript, authors introduced many complex material systems based on MOFs, and called these materials heterostructures. Here the question is what is the definition of MOFs-based heterostructures? Maybe the name of MOFs-based composites is better.

7.      In 3. Applications of MOFs and MOFs-based heterostructures, the organization of the reporting work needs to further strengthen the logic, rather than simply piling up the content.

8.      As mentioned in 5. Future outlook, the stability of MOFs greatly affects their reusability as photocatalysts, which is important for the topics covered in this review.  Therefore, it is hoped that more work to improve the stability of such frame materials will be mentioned in the text. One of the solutions is to in-situ growth of MOFs on woods. Please add this point and refer the following highly relevant works: MOFs meet wood: reusable magnetic hydrophilic composites toward efficient water treatment with super-high dye adsorption capacity at high dye concentration; When MOFs meet wood: From opportunities toward applications.

9.      Some figures need further modifications to provide better resolution and readability, especially the texts in the figures.

10.  Please check the entire manuscript carefully. There are still some spelling mistakes and grammar issues. Also, please check the references carefully to ensure complete information is included.

Author Response

Reviewer 2:

Comments and Suggestions for Authors

Based on the synthesis and application of MOFs and MOFs -based heterostructures photocatalyst, the manuscript comprehensively introduces the synthesis and modification methods of MOFs and MOFs -based heterostructures. The application and mechanism of the materials as photocatalysis are emphasized.  In addition, the manuscript collects and summarizes the literature related to the field comprehensively, and the mechanism of action is well explained through the Mechanism diagrams. However, there are still some issues to be addressed. A moderate revision is suggested before its acceptance.

  1. The introduction of MOFs as photocatalysts in the Introduction section emphasizes the advantages of 3D MOFs, but the structure and mechanism of 2D MOFs seem to be ignored.

Reply: Thank you. In this manuscript, we have covered both 2D and 3D MOFs for example, under sub-heading 2.1.1 and subheadings 3.

  1. More background on the photocatalysis should be provided with more supporting recent articles: Hydrothermal Synthesis of Ce-doped ZnO Heterojunction Supported on Carbon Nanofibers with High Visible Light Photocatalytic Activity; A short review on heterojunction photocatalysts: Carrier transfer behavior and photocatalytic mechanisms; etc. In addition, authors listed many applications of MOFs. One more important application of lithium storage should also be referred: Hierarchical porous Co3O4 nanocages with elaborate microstructures derived from ZIF-67 toward lithium storage.

Reply: Thank you for your suggestion. We have cited the mentioned articles in the appropriate places: on line 7 (page 1) and line 8 under subheading 2.2 (page 8). However, even though lithium storage is an important application, it does not fit in this review because we are focusing mostly on the photocatalysis area.

  1. Topological structure is an important aspect of the spatial structure of MOFs, and at the same time has an important impact on its performance. But this factor has not been mentioned in the article.

Reply: Thank you for your suggestion. We have included this factor under section 2.1, 1st paragraph, and lines 8-12.

  1. In 2.1. Synthesis of MOFs, the introduction of various methods of synthesis is slightly illogical, and it is recommended to follow a certain logical order, such as the synthesis conditions are harsh to simple, etc.

Reply: Thank you for your suggestion. We have categorized the synthesis methods accordingly on pages 5-8.

  1. How to construct the MOFs-based heterostructures are important. One more sub-section should be added into the manuscript.

Reply: Thank you for your suggestion. We have changed the subheading of 2.2. as a synthesis method of MOFs-based heterostructures where we have explained different modification methods on page 9.

  1. In the manuscript, authors introduced many complex material systems based on MOFs, and called these materials heterostructures. Here the question is what is the definition of MOFs-based heterostructures? Maybe the name of MOFs-based composites is better.

Reply: Thank you for your suggestion. The terms heterostructure and composite are not strictly divided. The heterostructure is also a generalized term for composite materials. Because of this, we will use the term heterostructure throughout the manuscript.

  1. In 3. Applications of MOFs and MOFs-based heterostructures, the organization of the reporting work needs to further strengthen the logic, rather than simply piling up the content.

Reply: Thank you for your suggestion. We have divided and reorganized the subsections in the application part.

  1. As mentioned in 5. Future outlook, the stability of MOFs greatly affects their reusability as photocatalysts, which is important for the topics covered in this review.  Therefore, it is hoped that more work to improve the stability of such frame materials will be mentioned in the text. One of the solutions is to in-situ growth of MOFs on woods. Please add this point and refer the following highly relevant works: MOFs meet wood: reusable magnetic hydrophilic composites toward efficient water treatment with super-high dye adsorption capacity at high dye concentration; When MOFs meet wood: From opportunities toward applications.

Reply: Thank you for your suggestion. We have added this point in the future outlooks on page 28.

  1. Some figures need further modifications to provide better resolution and readability, especially the texts in the figures.

Reply: Thank you for your comment. We have changed the texts in the figures especially for Figures 4 and 6 to make them more readable on pages 23 and 26 respectively.

  1. Please check the entire manuscript carefully. There are still some spelling mistakes and grammar issues. Also, please check the references carefully to ensure complete information is included.

Reply: Thank you for your comment. We have checked and revised the references accordingly.

Round 2

Reviewer 1 Report

Here are my comments on the manuscript titled Recent Progress of Metal-Organic Frameworks and Metal-Organic Frameworks-based heterostructures as Photocatalysts.

Decision: Accept

The authors have well addressed my concerns. Hence, I recommend publishing this work in Nanomaterials. Prior to this, please read thoroughly the references. I list them as following: wrong subscript of terms in Ref. 18, 32, 52, 53, 80, 97, 133, 141, 143, 145, 147, and 150. Ref. 53 is duplicated to Ref. 65.

Author Response

Reply: Thank you for your comments. We have revised the references as suggested.

Reviewer 2 Report

Authors have addressed most of issues well except some issues.

1. Authors have provided two email addresses for two authors. However, there are one corresponding authors and two coauthors. Please carefully read the journal instructions, if the email address for corresponding author or all authors should be provided.

2. The keywords should be modified. There are duplicated points in the keywords.

3. Authors should carefully treat the definition in the title for the "heterostructures". What kinds of structures in this review can be belonged to the scope of "heterostructures"? Maybe one more scheme should be provided to help for the definition and explaination.

4. In introduction, authors only introduced the background of photocatalysis and MOFs. However, half of the content is about the MOF-based heterostructures. Therefore, more background on the MOF-based heterostructures should be provided to have a better story line.

5. The future outlook section is too simple. All the points should be further modified to show why the points are challenge, and also should provide the possible solutions with supporting articles. Please refer to the previous comments 8 in first round reviewing report with supporting articles:  MOFs meet wood: reusable magnetic hydrophilic composites toward efficient water treatment with super-high dye adsorption capacity at high dye concentration; When MOFs meet wood: From opportunities toward applications. In addition, more outlook should be performed for the MOF-bases heterostructures.

6. The conclusion section is more likely an abstract and it is also repeated with the last paragraph of introduction section. Authors can refer to some high impact review article and rewrite this section.

Author Response

Reviewer 2:

Comments and Suggestions for Authors

Authors have addressed most of issues well except some issues.

  1. Authors have provided two email addresses for two authors. However, there are one corresponding authors and two coauthors. Please carefully read the journal instructions, if the email address for corresponding author or all authors should be provided.

Reply: Thank you for your comments. Both email addresses are the corresponding author’s email IDs. Based on the journal instruction, we are asked to provide the corresponding author’s email ID only.

  1. The keywords should be modified. There are duplicated points in the keywords.

Reply: Thank you for your comments. The keywords have been revised (on page 1).

  1. Authors should carefully treat the definition in the title for the "heterostructures". What kinds of structures in this review can be belonged to the scope of "heterostructures"? Maybe one more scheme should be provided to help for the definition and explanation.

Reply: Thank you for your comment and suggestions. The definition of the MOF-based heterostructures has been added on pages 4-5. The different schemes can be seen in Figure 6 (on page 25) and Figure 7 (on page 26).

  1. In introduction, authors only introduced the background of photocatalysis and MOFs. However, half of the content is about the MOF-based heterostructures. Therefore, more background on the MOF-based heterostructures should be provided to have a better story line.

Reply: Thank you for your comment and suggestions. The background on the MOF-based heterostructures has been added on pages 4-5.

  1. The future outlook section is too simple. All the points should be further modified to show why the points are challenge, and also should provide the possible solutions with supporting articles. Please refer to the previous comments 8 in first round reviewing report with supporting articles:  MOFs meet wood: reusable magnetic hydrophilic composites toward efficient water treatment with super-high dye adsorption capacity at high dye concentration; When MOFs meet wood: From opportunities toward applications. In addition, more outlook should be performed for the MOF-bases heterostructures.

Reply: Thank you for your comments. We have added supporting articles in the future outlook.

  1. The conclusion section is more likely an abstract and it is also repeated with the last paragraph of introduction section. Authors can refer to some high impact review article and rewrite this section.

Reply: Thank you for your comments and suggestion. The conclusion has been revised (page 30).

Round 3

Reviewer 2 Report

Authors have addressed all the issues. An acceptance is suggested.